# The profile of lipid metabolites in urine of marmoset wasting syndrome

Arisa Yamazaki[1☯], Tatsuro Nakamura[1☯], Takako Miyabe-Nishiwaki[2], Akihiro Hirata[3], Rikako Inoue[1], Koji Kobayashi[1], Yusuke Miyazaki[1], Yuta Hamasaki[1], Akiyo Ishigami[2], Nanae Nagata[1], Akihisa Kaneko[2], Makoto Koizumi[4], Hiroki Ohta[5], Hirotaka James Okano[5], Takahisa Murata[1]*

1 Department of Animal Radiology, Graduate School of Agricultural and Life Sciences, The University of Tokyo, Tokyo, Japan, 2 Department of Primate Research Institute, Kyoto University, Aichi, Japan, 3 Department of Animal Experiment, Life Science Research Center, Gifu University, Gifu, Japan, 4 Department of Laboratory Animal Facilities, The Jikei University School of Medicine, Tokyo, Japan, 5 Department of Regenerative Medicine, The Jikei University School of Medicine, Tokyo, Japan

☯ These authors contributed equally to this work.
* amurata@mail.ecc.u-tokyo.ac.jp

## Abstract

Marmoset wasting syndrome (MWS) is clinically characterized by progressive weight loss. Although morbidity and mortality of MWS are relatively high in captive marmosets, its causes remain unknown. Lipid mediators are bioactive metabolites which are produced from polyunsaturated fatty acids, such as arachidonic acid (AA) and eicosapentaenoic acid. These lipid metabolites regulate a wide range of inflammatory responses and they are excreted into the urine. As urinary lipid profiles reflect systemic inflammatory conditions, we comprehensively measured the levels of 141 types of lipid metabolites in the urines obtained from healthy common marmoset (*Callithrix jacchus*) (N = 7) or marmosets with MWS (N = 7). We found that 41 types of metabolites were detected in all urine samples of both groups. Among them, AA-derived metabolites accounted for 63% (26/41 types) of all detected metabolites. Notably, the levels of AA-derived prostaglandin (PG) $E_2$, $PGF_{2\alpha}$, thromboxane (TX) $B_2$ and $F_2$-isoprostanes significantly increased in the urine samples of marmosets with MWS. In this study, we found some urinary lipid metabolites which may be involved in the development of MWS. Although the cause of MWS remains unclear, our findings may provide some insight into understanding the mechanisms of development of MWS.

## Introduction

Marmoset wasting syndrome (MWS) is one of the leading causes of morbidity and mortality in captive marmosets [1]. The common clinical signs of MWS are chronic weight loss, diarrhea and anemia [2]. Although several different lesions have been pathologically found in marmosets with clinical signs of MWS [3, 4], chronic colitis is considered one of the most important contributing factors for the development of MWS [5–7]. Tranexamic acid and glucocorticoid are often used to treat symptoms [1, 8]. The other options, such as dietary manipulation and

**Data Availability Statement:** All relevant data are within the manuscript and its Supporting Information files.

**Funding:** This work was supported by the Cooperative Research Program of Primate Research Institute, Kyoto University, the Japan Society for the Promotion of Science, the Takeda Science Foundation, the Nipponham Foundation, Hoyu Science Foundation, and the Futaba Foundation to T.M.

**Competing interests:** The authors have declared that no competing interests exist.

stress minimization, also may help reduce the risks of developing MWS [2]. However, there is no reliable treatment for MWS. There is also no definitive antemortem diagnosis method for MWS.

Polyunsaturated free fatty acids (PUFAs) are divided into omega-6 (*n*-6) and omega-3 (*n*-3) groups based on double bond positions. The *n*-6 PUFAs include linoleic acid (LA), dihomo-γ-linolenic acid (DGLA) and arachidonic acid (AA), whereas *n*-3 PUFAs include α-linoleic acid (ALA), eicosapentaenoic acid (EPA) and docosahexaenoic acid (DHA). Three types of oxygenases, cyclooxygenase (COX), lipoxygenase (LOX) and cytochrome P450 (CYP) epoxygenase oxidize PUFAs and convert to lipid mediators and/or their related metabolites [9].

Lipid mediators are bioactive molecules and have identified hundreds of different types in the mammalian body. It has been demonstrated that lipid mediators positively and/or negatively regulate inflammatory responses [10]. For example, AA-derived COX metabolite prostaglandin (PG) $E_2$ ($PGE_2$) induces arterial dilation and vascular hyper-permeability resulting in edema, which is a classic sign of inflammation [11]. Another AA-derived pro-inflammatory mediator leukotriene $B_4$ induces chemotaxis and the production of pro-inflammatory cytokines in neutrophils [12, 13]. In contrast, the EPA-derived LOX metabolite resolvin exerts anti-inflammatory properties by inhibiting the recruitment of leukocytes and/or the production of pro-inflammatory cytokines [14].

Lipid mediators and their related metabolites are present in the urine and their levels change according to the host inflammatory condition. Indeed, some specific lipid metabolites positively correlate with disease activity score in human rheumatoid arthritis [15]. In addition, our previous study demonstrated that a major urinary metabolite of $PGD_2$, tetranor-PGDM, positively correlates with the severity of food allergies in mice and humans [16]. Thus, the profiles of urinary lipid metabolites can be convenient indices of various types of diseases.

On the basis of these backgrounds, we performed an LC-MS-based comprehensive analysis of lipid metabolites in the urines of marmosets suffering from MWS to understand pathophysiological features and to provide some insight for the development of a diagnostic method.

## Materials and methods

### Humane care guidelines

All procedures used in this study followed the Guidelines for the Care and Use of Nonhuman Primates, provided by Primate Research Institute, Kyoto University (KUPRI). The protocol for urinary sampling was approved by Institutional Animal Care and Use Committee in KUPRI (approval number: 2018–079). We did not anesthetize or euthanize animals for the purpose of this study. No specific animal research protocol was drafted for this study in the Jikei University School of Medicine (Jikei) and Marmoset Breeding Facility, CLEA Japan, Inc. study as only clinical samples were provided to the study.

### Rearing conditions of marmosets

The marmosets were housed in family cages (W700 x D700 x H1500 mm or W910 x D700 x H1600 mm in KUPRI, W800 x D650 x H1500 mm in Jikei), paired cages (W1200 x D600 x H1000 mm in KUPRI, W800 x D650 x H720 mm in Jikei.) or individual/intensive care cages (W600 x D600 x H1000 mm in KUPRI, W400 x D650 x H720 mm in Jikei, W390 x D550 x H700 mm in CLEA). The room temperature was kept at 28 ± 5˚C. Water was available ad libitum. They were fed on 50 ml (20–30 g) pellet (SPS, Oriental Yeast Co. ltd., Tokyo, Japan) twice a day supplemented with apple and quail's egg three times a week, banana twice a week and occasional mealworm (the larva of Tenebrionidae) in KUPRI. In Jikei, they were fed on 40 g pellet (LabDiet, PMI Nutrition International LLC, U.S.A) twice a day supplemented with

**Table 1. The concentration of PUFAs contained in diet.**

| | PUFAs (g/100 g food) | KUPRI | Jikei | CLEA |
|---|---|---|---|---|
| n-6 | Linoleic acid | 4.16 (1.66–2.82) | 3.12 (2.5) | 4.70 (1.88) |
| | Arachidonic acid | N.D. | 0.04 (0.03) | N.D. |
| n-3 | Linolenic acid | 0.24 (0.10–0.32) | 0.31 (0.25) | 0.54 (0.21) |
| | DHA | 0.11 (0.02–0.05) | 0.35 (0.28) | 0.04 (0.02) |
| | EPA | 0.08 (0.01–0.03) | | 0.02 (0.01) |

The values in parentheses indicate the ingested concentration of PUFAs per day. KUPRI, Kyoto University Primate Research Institute; Jikei, The Jikei University School of Medicine; CLEA, Marmoset Breeding Facility, CLEA Japan, Inc.

honey, Vitamin C and D. In CLEA, they were fed on 40–50 g CMS-1M (CLEA Japan, Inc., Tokyo, Japan) soaked in water and supplemented with Vitamin C and D once a day, and boiled eggs twice a week. Estimated concentrations of n-6 and n-3 PUFAs contained in diet were comparable in each institution or between healthy and MWS marmosets (Table 1). In all institutions, environmental enrichment, including gum feeders, wooden toys, climbing structures and swings, was provided depending on the housing condition. The animal care staffs observed the health and well-being of the animals daily using criteria, such as fecal condition, appetite, hair condition and movement, and if they were suspected to have any problems, they would be evaluated by the veterinarians. After the study, control marmosets were used in different projects and MWS marmosets underwent further treatment.

## Identification of MWS

The major criteria of MWS were based on the previous report; recurrent diarrhea, absence of pathogenic enteric bacteria and endoparasites, resolution of illness despite anti parasitic treatment and/or antibiotics, and persistent weight loss [2]. The veterinarians in each institution assessed the animals prior to inclusion in the study. The information of each marmoset was shown in the Table 2. In general, healthy adult marmosets weigh around 350 g in captivity

**Table 2. Body weight and clinical signs in individual animals.**

| | No. | Institution | Gender | Age (year) | Maximum BW (g) | BW (g) | Clinical signs |
|---|---|---|---|---|---|---|---|
| Healthy | 1 | KUPRI | male | 4 | 370 | 354 | None |
| | 2 | KUPRI | male | 6 | 384 | 378 | None |
| | 3 | Jikei | male | 9 | 323 | 320 | None |
| | 4 | CLEA | female | 1 | 290 | 290 | None |
| | 5 | CLEA | male | 2 | 420 | 375 | None |
| | 6 | CLEA | male | 1 | 340 | 335 | None |
| | 7 | CLEA | male | 2 | 310 | 310 | None |
| MWS | 1 | KUPRI | male | 6 | 336 | 280 | frequent, recurring diarrhea, persistent weight loss |
| | 2 | Jikei | male | 4 | 350 | 193 | persistent weight loss |
| | 3 | CLEA | male | 3 | 265 | 240 | recurring diarrhea, persistent weight loss |
| | 4 | CLEA | male | 9 | 350 | 225 | recurring diarrhea, persistent weight loss |
| | 5 | CLEA | male | 6 | 345 | 300 | recurring diarrhea, persistent weight loss |
| | 6 | CLEA | female | 7 | 340 | 290 | recurring diarrhea, persistent weight loss |
| | 7 | CLEA | female | 5 | 340 | 215 | recurring diarrhea, persistent weight loss |

KUPRI, Kyoto University Primate Research Institute; Jikei, The Jikei University School of Medicine; CLEA, Marmoset Breeding Facility, CLEA Japan, Inc.

[17]. In this study, the body weight of each healthy marmoset exceeded 300 g (except for No.4, which was 1 year old and still growing), however, that of marmosets with MWS were around or less than 300 g at urine samples collection. Frequent and recurrent diarrhea, and persistent weight loss for over six months were observed in marmosets with MWS. Marmosets with MWS had a history of medication, such as ursodeoxycholic acid and pancrelipase, but which had been taken approximately 1 month before urine samples collection.

## Urine samples

The urine samples were collected from common marmosets (*Callithrix jacchus*) diagnosed as healthy (N = 7) in primary care or MWS (N = 7). Some of the control marmosets were in pair or family cages, while others were in individual cages for research or management purpose (not related to this study). MWS marmosets were in individual cages when their conditions were deteriorating and required intensive care [2]. When the marmoset was in a pair or family cage, the marmoset was temporarily separated from other individuals using separation wall to collect urine samples. In some occasions, even in a pair or family cage, when a marmoset was urinated in front of us, the urine sample was collected immediately. A clean dry tray was placed under the cage. All urine samples were collected only when visually confirmed not to be contaminated with feces, another individual's urine, or dropped food. After each sample collection, the trays were sterilized, rinsed thoroughly and dried. The urine samples were stored until use at -80˚C.

## Sample preparation

After the urine samples were centrifuged at 20,000 x *g* for 5 min, aliquot of 200 μl urine was mixed with 350 μl of 0.1% formic acid water and 50 μl of the internal standard solution. The mixed solutions were applied to solid phase extraction cartridge (OASIS μElution plate, Waters, Massachusetts, USA) preconditioned with 200 μl methanol and distilled water (DW). After washing with 200 μl DW and 200 μl hexane, lipids fractions were eluted with 100 μl methanol.

## LC-MS-based comprehensive analysis of lipid metabolites

The 5 μl sample solution was injected to high performance liquid chromatography (Nexera 2, Shimadzu, Kyoto, Japan) equipped with mass spectrometer (LCMS-8060, Shimadzu, Kyoto, Japan). 138 types of metabolites, 3 types of PUFAs and 15 types of internal standards were measured and analyzed by using LC/MSMS Method Package for Lipid Mediators version 2 with LabSolutions software (Shimadzu, Kyoto, Japan) as manufacturers instruction (S1 Table). Each metabolite was identified by retention time and selected reaction monitoring ion transition (S1 Table). The change of each metabolite level between healthy and MWS urines was examined by comparing the peak area ratio calculated as following formula; The peak area of each metabolite / the peak area of internal standard. Each value was further corrected by the concentration of creatinine measured by LabAssay™ Creatinine (Wako, Osaka, Japan).

## Statistical analysis

All data are shown as mean ± SEM. The statistical difference was determined by Mann-Whitney U test for two-group comparison. Statistically significance was determined when p-value is less than 0.05.

## Results

### Detected lipid metabolites in the urines of MWS

We measured 141 types of lipid metabolites including 3 types of PUFAs (AA, EPA and DHA) using urine samples from healthy marmoset (N = 7) and marmosets with MWS (N = 7) (Table 2 and S1 Table). Marmosets in each institution were fed different food, but marmosets with heathy and MWS were fed the same food in each institution (Table 1). In addition, to unveil the inflammatory features of MWS, we focused and analyzed the metabolites that were constantly detected in all MWS individuals. Under these criteria, we detected 3 types of PUFAs (Fig 1) and 38 types of metabolites in total (Figs 2–4). In the detected 38 types of metabolites, $n$-6 PUFAs-derived metabolites accounted for 82% (31/38 types), on the other hand, the rest of metabolites were consisted of $n$-3 PUFAs-derived metabolites (7/38 types). Furthermore, AA-derived metabolites accounted for 66% (25/38 types) of total detected metabolites. The amount of AA tended to decrease but that EPA or DHA did not change in MWS urines compared to healthy urines (Fig 1). These results suggest that $n$-6 fatty acids, especially AA, are mainly consumed in MWS marmosets.

### Metabolic pathways of detected lipid metabolites

The metabolic pathways of all detected metabolites were drawn in Figs 2, 3 and 4. To determine which lipid mediators are critically involved in the development of MWS, we focused on the lipid metabolites that significantly increased in the urines of MWS than that of healthy marmosets. As shown in Fig 2, the major AA metabolites, $PGE_2$, $PGF_{2\alpha}$ and its metabolite of 13, 14-dihydro-15-keto-tetranor-$PGF_{2\alpha}$ and a metabolite of $TXA_2$, $TXB_2$ were detected at significant higher levels in the urines of MWS than that of healthy marmosets. These results suggest that those three types of mediators may play important roles in the development and progression of MWS (Fig 2).

Isoprostanes are non-enzymatic oxidative products of PUFAs. $F_2$-isprostanes ($F_2$-isoPs) are prostaglandin $F_2$-like isoprostanes generated from AA. In this study, 5 types of $F_2$-isoPs were detected and 4 types of that were significantly increased in MWS urines (Fig 3). Several studies reported that the amount of $F_2$-isoPs were elevated under oxidative stress upon inflammation [18, 19]. Thus, detection of these $F_2$-isoPs in urine may evidence the presence of inflammation associated oxidative stress in MWS.

In addition, DGLA-derived $F_1$-isoprostane ($F_1$-IsoP), 8-iso-$PGE_1$, also significantly increased in the urine samples of MWS (Fig 4). The other PUFAs-derived urinary metabolites did not change between healthy and MWS group (Fig 4). It is reported that $F_1$-IsoPs also produced by oxidative stress, however, the usefulness of urinary $F_1$-IsoP as biomarker has not yet been reported. 8-iso-$PGE_1$ may be a unique metabolite increased in MWS urine.

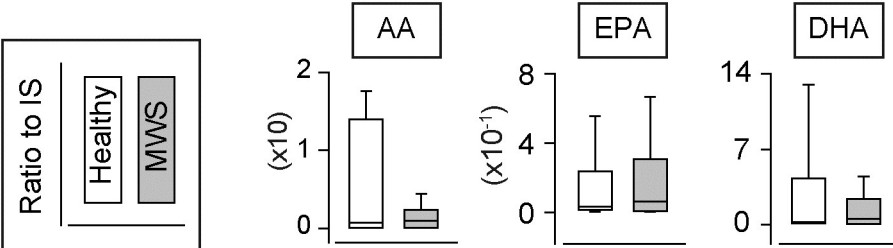

**Fig 1. The urinary level of PUFAs.** The urinary levels of arachidonic acid (AA), eicosapentaenoic acid (EPA) and docosahexaenoic acid (DHA) in healthy marmoset (N = 7) and marmosets with MWS (N = 7).

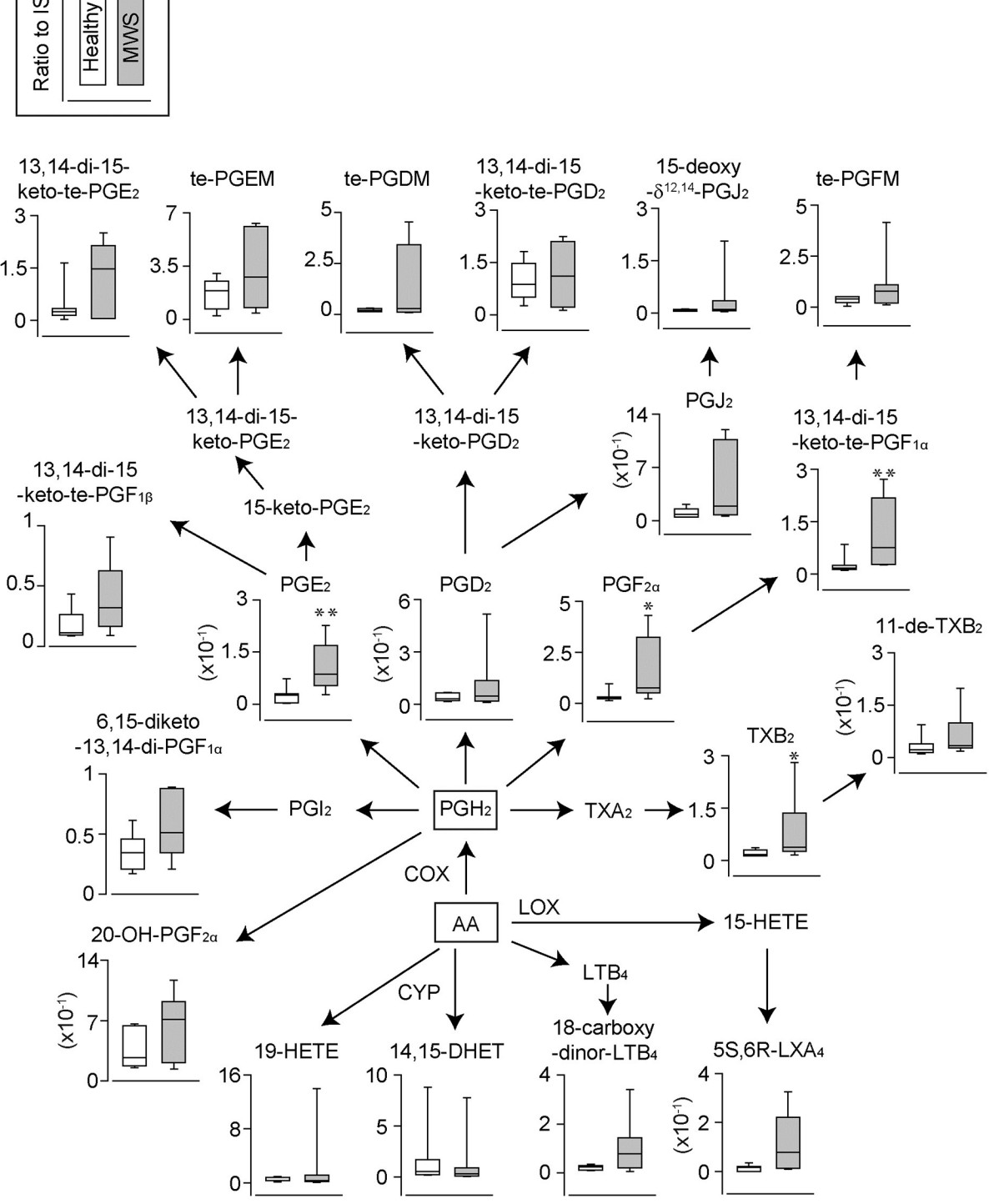

**Fig 2. The urinary levels of AA-derived enzymic-oxidative metabolites.** The urinary levels of arachidonic acid (AA)-derived catalyzed metabolites in healthy marmoset (N = 7) and marmosets with MWS (N = 7). di. dihydro; te, tetranor; COX, Cyclooxygenase; CYP, cytochrome P450 epoxygenase; LOX, lipoxygenase. $^*$p<0.05, $^{**}$p<0.01 compared to healthy urines.

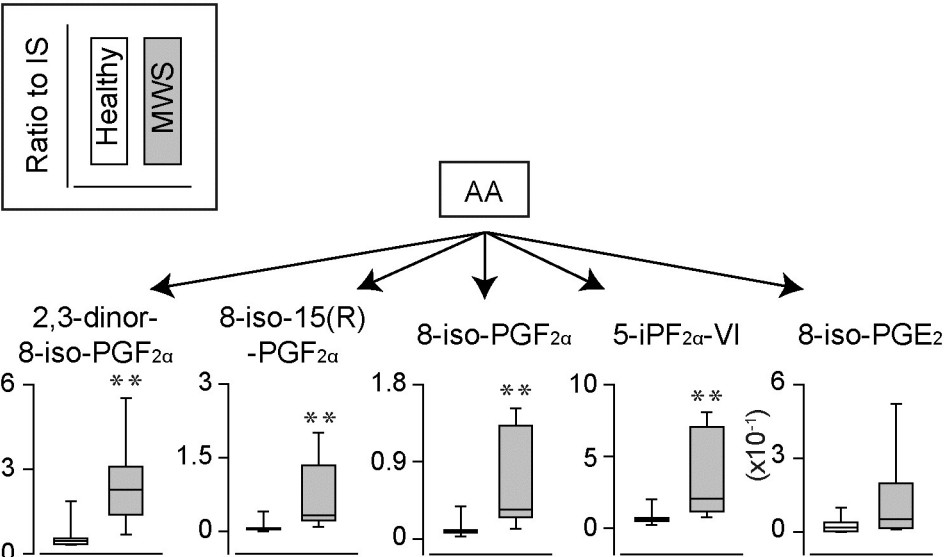

**Fig 3. The urinary levels of AA-derived isoprostanes.** The urinary levels of arachidonic acid (AA)-derived isoprostanes produced by non-enzymatic oxidation in healthy marmoset (N = 7) and marmosets with MWS (N = 7). **p<0.01 compared to healthy urines.

## Discussion

Several stimuli immediately activate appropriate oxidases, such as COX and LOX, and lead *de novo* synthesis of lipid mediators from *n*-6 and/or *n*-3 PUFAs. Generally, *n*-6 PUFA AA-derived lipid mediators accelerate inflammation pathways in both the acute and chronic phases. On the other hand, *n*-3 PUFAs EPA- and DHA-derived mediators accelerate the resolution of inflammation at the chronic phase [20, 21]. Thus, the number, types, and levels of lipid mediators are changed according to the severity and/or duration of inflammation. The urinary lipid profiling reflects these changes in host inflammatory condition as the dynamic balance between *n*-6 and *n*-3 PUFAs-derived metabolites. In the present study, *n*-6 PUFA, especially AA, -derived metabolites accounted for a majority of urinary lipid metabolites in marmosets with MWS. These results may reflect that MWS marmosets may have persistent inflammation accompanied by AA consumption.

Even though $PGE_2$ and $PGF_{2\alpha}$ are highly metabolically unstable, these compounds were detected in the urines of marmoset. In addition to $PGE_2$ and $PGF_{2\alpha}$, primary metabolites of these PGs in plasma, 13,14-dihydro-15-keto derivatives, were increased in the urines of marmosets with MWS. [22]. Thus, a large amount of these PGs would be produced in the inflammatory lesions and excreted into the MWS urines. However, previous studies have shown that some of the urinary $PGE_2$ or $PGF_{2\alpha}$ are formed by free-radical catalyzed oxidation pathway (IsoP pathway) in the rodent and human [23, 24]. It cannot be ignored the possibility that urinary $PGE_2$ and/or $PGF_{2\alpha}$ were generated by IsoP pathway and the increase of their levels reflect a free radical production by oxidative stress.

Chronic enteritis is a major post-mortal pathological feature of MWS [5–7]. We have also found many cases with severe chronic enteritis in marmosets clinically diagnosed with MWS in the colony at the KUPRI. In the present study, we found that the amounts of $PGE_2$, $PGF_{2\alpha}$ and a metabolite or $TXA_2$ were markedly increased in the urines of marmosets with MWS. Clinical studies have shown that $PGE_2$ levels are elevated in the urine of patients with intestinal inflammation [25]. Gene deficiency or pharmacological blocking of the $PGE_2$ receptor, EP4,

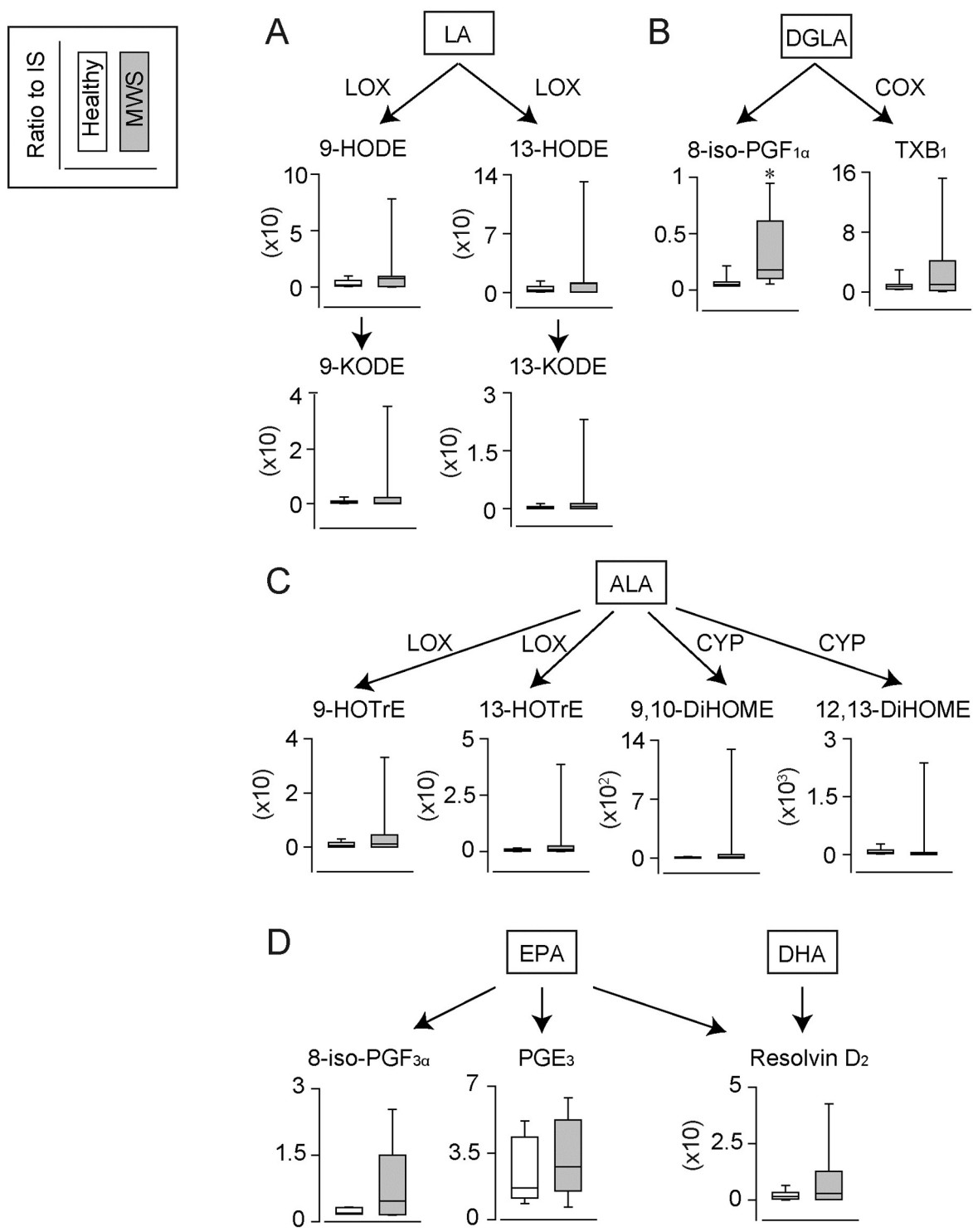

**Fig 4. The urinary levels of lipid metabolites derived from PUFAs except AA.** The urinary levels of (A) linoleic acid (LA), (B) dihomo-γ-linolenic acid (DGLA), (C) α-linoleic acid (ALA), (D) eicosapentaenoic acid (EPA) and (E) docosahexaenoic acid (DHA)-derived metabolite in healthy marmoset (N = 7) and marmosets with MWS (N = 7). *p<0.05 compared to healthy urines. LOX, lipoxygenase; COX, cyclooxygenase; CYP, cytochrome P450 epoxygenase.

ameliorated dextran sodium sulfate–induced colitis in mice [26]. Collins D et al. showed in an *in vitro* study that $PGF_{2\alpha}$ promoted chloride secretion in human colonic epithelial cells through a cAMP-mediated mechanism [27]. Further, $TXA_2$ up-regulates neutrophil elastase release and aggravate trinitrobenzene sulfonic acid-induced guinea-pig colitis [28]. Our findings and those reports suggest that $PGE_2$, $PGF_{2\alpha}$ and $TXA_2$ may play important roles in the development and progression of MWS. Further investigations are needed to reveal the functional relevance of these mediators in MWS.

In this study, we found that $F_2$-IsoPs were increased in the urine samples of marmosets with MWS. It is well known that the levels of urinary $F_2$-IsoPs elevated in the presence of oxidative stress. Indeed, urinary levels of $F_2$-isoP is elevated in the patients with Chron's diseases [29]. Considering that isoprostanes are chemically stable, urinary $F_2$-IsoPs may be urinary biomarker of MWS. In addition to $F_2$-IsoPs, we also found the increased level of $F_1$-isoP, 8-iso $PGF_{1\alpha}$, which possibly be unique in MWS urine samples. To demonstrate the relevance of the urinary levels of $F_2$-isoPs and $F_1$-IsoP to the onset of MWS may be able to establish a new early diagnosis method for MWS.

The *n*-3 PUFAs are essential fatty acids that are identified as anti-inflammatory modulators by inhibiting AA metabolism and/or producing mediators, such as resolvin, lipoxin and protectin [30]. The decreased levels of *n*-3 PUFAs were reported in patients with inflammatory bowel disease. In this study, only 5 types of *n*-3 PUFAs-derived metabolites were constantly detected in the urine samples of marmosets with MWS. Additionally, the levels of *n*-3 PUFAs did not change in both groups. These results raised the possibility that *n*-3 PUFAs are not metabolized in MWS marmoset. Several reports have suggested that supplementation with *n*-3 PUFAs improved inflammation, including inflammatory bowel disease [31, 32]. The feeding of *n*-3 PUFAs containing food may be effective for the treatment of MWS.

To our knowledge, this is the first study assessing the urinary lipid profile of MWS. We found some lipid metabolites which may be involved in the development of MWS. Since MWS is a multifactorial disease and the cause of it remains unclear, we need further investigation to reveal a causal relationship. We believe our findings may provide some insight into understanding the mechanisms of development of MWS.

## Supporting information

**S1 Table. Measured 141 types of lipid metabolites and 15 types of internal standards.** PUFA, polyunsaturated free fatty acid; SRM, selected reaction monitoring; ALA, α-linoleic acid; LA, linoleic acid; DGLA, dihomo-γ-linolenic acid; AA, arachidonic acid; EPA, eicosapentaenoic acid; DHA, docosahexaenoic acid.
(DOCX)

## Acknowledgments

The authors thank Dr. Satoshi Maeda and Dr. Rie Nishii (Marmoset Breeding Facility, CLEA Japan, Inc, Gifu, Japan) for providing urine samples.

## Author Contributions

**Conceptualization:** Takahisa Murata.

**Data curation:** Arisa Yamazaki, Tatsuro Nakamura, Takako Miyabe-Nishiwaki, Akihiro Hirata, Rikako Inoue, Koji Kobayashi, Yusuke Miyazaki, Yuta Hamasaki, Akiyo Ishigami, Nanae Nagata, Akihisa Kaneko, Makoto Koizumi, Hiroki Ohta, Hirotaka James Okano.

**Formal analysis:** Arisa Yamazaki, Tatsuro Nakamura, Koji Kobayashi.

**Funding acquisition:** Takahisa Murata.

**Investigation:** Arisa Yamazaki, Takahisa Murata.

**Methodology:** Tatsuro Nakamura, Takako Miyabe-Nishiwaki, Koji Kobayashi, Takahisa Murata.

**Project administration:** Tatsuro Nakamura, Takahisa Murata.

**Resources:** Hiroki Ohta, Hirotaka James Okano, Takahisa Murata.

**Supervision:** Takahisa Murata.

**Validation:** Arisa Yamazaki.

**Writing – original draft:** Arisa Yamazaki, Tatsuro Nakamura, Takahisa Murata.

**Writing – review & editing:** Takako Miyabe-Nishiwaki.

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
