## [Decision Letter · Decision Letter 0]

16 Mar 2020

PONE-D-20-02190

The production profile of lipid metabolites in urine of marmoset with wasting syndrome

PLOS ONE

Dear Prof. Murata,

Thank you for submitting your manuscript to PLOS ONE. After careful consideration, we feel that it has merit but does not fully meet PLOS ONE’s publication criteria as it currently stands. Therefore, we invite you to submit a revised version of the manuscript that addresses the points raised during the review process. Your manuscript was received with mixed opinions. When preparing the manuscript revision, please pay special attention to comments of reviewer #3, some of which may be difficult to address or may actually prevent the manuscript publication at all.

We would appreciate receiving your revised manuscript by Apr 24 2020 11:59PM. To enhance the reproducibility of your results, we recommend that if applicable you deposit your laboratory protocols in protocols.io, where a protocol can be assigned its own identifier (DOI) such that it can be cited independently in the future. For instructions see: http://journals.plos.org/plosone/s/submission-guidelines#loc-laboratory-protocols

We look forward to receiving your revised manuscript.

Kind regards,

Petr Heneberg

Academic Editor

PLOS ONE

Journal Requirements:

"This work was supported by the Cooperative Research Program of Primate Research Institute, Kyoto University, the Japan Society for the Promotion of Science, the Takeda Science Foundation, the Nipponham Foundation, Hoyu Science Foundation, and the Futaba Foundation to T.M.."

We note that one or more of the authors are employed by a commercial company: CLEA Japan.

Reviewers' comments:

Reviewer's Responses to Questions

**Comments to the Author**

1. Is the manuscript technically sound, and do the data support the conclusions?

Reviewer #1: Yes

Reviewer #2: Partly

Reviewer #3: No

2. Has the statistical analysis been performed appropriately and rigorously? 

Reviewer #1: Yes

Reviewer #2: Yes

Reviewer #3: No

3. Have the authors made all data underlying the findings in their manuscript fully available?

Reviewer #1: Yes

Reviewer #2: No

Reviewer #3: No

4. Is the manuscript presented in an intelligible fashion and written in standard English?

Reviewer #1: Yes

Reviewer #2: Yes

Reviewer #3: Yes

5. Review Comments to the Author

Reviewer #1: This work deals with PUFA selected metabolites in Marmosets suffering MWS. Because the cause is still unknown the subject should be deemed quite interesting. By comparison to the attached comments to the previous reviews I find the work almost acceptable for PLoS ONE. Unfortunately, we still do not know what is the real cause of MWS but the authors provided some mediators and mechanism for the development of that illness.

Some minor remarks:

- Please add clearly in the conclusion that the cause of MWS remains unknown but the paper revealed some mechanisms and mediators of MWS

- Lines 79-83: the authors provide information about different types of cages used in the experiment. Please make it clear for the reader when and why such cages were used. I suppose that an individual cage is ideal for urine sampling. If another cages were used, how the urine was sampled? Add appropriate info sentence.

Reviewer #2: Thank you for your interest in MWS, not enough researches are focused on this to find a solution.

As much as I enjoyed your paper, I have a number of reservations that need to be addressed before I can recommend it for publishing. Find the notes added to the manuscript for details.

In large:

1) Diarrhoea itself does not equal MWS. You need to add a section on how you identified MWS. From the read of your paper, it sounds like you can't say you were looking at MWS but maybe focus on inflammation or chronic diarrhoea. If they do have MWS maybe you can present weight loss data.

2) You could do a better literature search, recent papers on MWS have not been included - see text for reference.

3) Can you give us the nutrient concentrations of their diet? How much AA, DHA and EPA are they actually ingesting?

4) as it stands, all your results support is that there is inflammation in animals with diarrhoea. You need the diet information to link with AA. If they eat the same diet and same amount of fatty acids, then causation will be somewhere else. You need the diet information to tie this all together, or change the scope of your paper to inflammation and darrhoea.

Reviewer #3: I commend the author's efforts to try and shed light on to disease like WMS which remains a mystery even after so many attempts, however I do think there are some major flaws in the design. I do understand that sample sizes can be challenging but in a prevalent colony as described by the owners, a n of 7 for both groups do significantly limit the conclusions that can be drawn.

The title says production profile - I would remove production.

I do think there are many authors, actually more authors than study subjects and it would be good to know if they satisfied the criteria for authorship.

Ln24: It is a syndrome and hence more symptoms are needed to justify the syndrome.

Ln 34: I do not think given the low number of subjects, a substantial conclusion can be got from this.

Ln 42: I do think it is redundant to have steroid and glucocorticoids in the same line

Ln 72: Sampling method should be elaborated. LN 104; Were there liners? How did you clean them in between. If feces and urine fall on the same surface,how does one confirm there is no residual effect of the feces. How do you know these inflammatory lipids were not from a biofilm on the floor? The trays needs to be described?

ln 84 onwards: Diet is a huge component of serum and urine lipid profiles. They should have looked at animals on the same diet to see how they compared.

Ln 104: Urines should be replaced with urine samples. What was the duration between voiding and urine collection and storage. Volatile lipids can potentially be lost.Marmosets in certain facilities are trained to urinate, was an attempt for this made

LN 111 onwards:What was the normalization technique used?Was the quantity of metabolites normalized to USG or creatinine.

There should also be some way to look into if there was active sediment or not, or at least clinical signs or something to substantiate that these mediators were not produced within the urinary tract as a result of inflammation or infection. WMS is syndrome and hence this is important.

ln 146: This is an over stretch and this conclusion is not substantiated.

I do agree with the other reviewers that the CV and the study design and conclusions have severe limitations,this along with small number of marmosets make me question the validity of these findings. While I do understand how difficult it is to work with marmosets, and samples can be scarce, I do not think the current study can be published as isin good faith.Sorry

182- urines should be replaced with urine samples.

6. PLOS authors have the option to publish the peer review history of their article (what does this mean?). If published, this will include your full peer review and any attached files.

Reviewer #1: No

Reviewer #2: Yes: Francis Cabana, PhD

Reviewer #3: Yes: Joseph Cyrus Parambeth

---

## [Author Response · Author response to Decision Letter 0]

18 Apr 2020

The production profile of lipid metabolites in urine of marmoset wasting syndrome. 

Review Comments to the Author

Reviewer #1: This work deals with PUFA selected metabolites in Marmosets suffering MWS. Because the cause is still unknown the subject should be deemed quite interesting. By comparison to the attached comments to the previous reviews I find the work almost acceptable for PLoS ONE. Unfortunately, we still do not know what is the real cause of MWS but the authors provided some mediators and mechanism for the development of that illness.

Comment 1: Please add clearly in the conclusion that the cause of MWS remains unknown but the paper revealed some mechanisms and mediators of MWS

<Response>

As the reviewer suggested, we added the descriptions in Abstract and manuscript in line 257 as following; Abstract; “In this study, we found some urinary lipid metabolites suspected to be involved in the development of MWS. Although the cause of MWS remains unclear, our findings may provide some insight into understanding the mechanisms of development of MWS.”

line 257; “We found some lipid metabolites which may be involved in the development of MWS. Since MWS is multifactorial disease and the cause of it remains unclear, we need further investigation to reveal a causal relationship. We believe our findings may provide some insight into understanding the mechanisms of development of MWS.” 

Comment 2: Lines 79-83: the authors provide information about different types of cages used in the experiment. Please make it clear for the reader when and why such cages were used. I suppose that an individual cage is ideal for urine sampling. If another cages were used, how the urine was sampled? Add appropriate info sentence.

<Response>

Some of the control marmosets were in pair or family cages, while others were in individual cages for research or management purpose (not related to this study). MWS marmosets were in individual cages when their conditions were deteriorating and required intensive care. When the marmoset was in a pair or family cage, the marmoset was temporarily separated from other individuals using separation wall to collect urine samples. In some occasions, even in a pair or family cage, when a marmoset was urinated in front of us, the urine sample was collected immediately. A clean dry tray was placed under the cage. All urine samples were collected only when visually confirmed not to be contaminated with feces, another individual’s urine, or dropped food. After each sample collection, the trays were sterilized, rinsed thoroughly and dried. The urine samples were stored until use at -80°C. We added the descriptions in Methods “Urine Samples” section in line 116.

Reviewer #2: Thank you for your interest in MWS, not enough researches are focused on this to find a solution. As much as I enjoyed your paper, I have a number of reservations that need to be addressed before I can recommend it for publishing. Find the notes added to the manuscript for details.

Comment 1: Diarrhea itself does not equal MWS. You need to add a section on how you identified MWS. From the read of your paper, it sounds like you can't say you were looking at MWS but maybe focus on inflammation or chronic diarrhea. If they do have MWS maybe you can present weight loss data.

<Response> 

We identified MWS by the major criteria based on Cabana et al 2018; recurrent diarrhea, absence of pathogenic enteric bacteria and endoparasites, resolution of illness despite anti parasitic treatment and/or antibiotics, and persistent weight loss. The veterinarians in each institution assessed the animals and included in this study. All individuals with MWS had persistent weight loss for over six months. We added the descriptions in “Identification of MWS” in Methods section and the maximum body weight in Table 2.

Comment 2: You could do a better literature search, recent papers on MWS have not been included see text for reference.

<Response> We added the paper “Francis Cabana et al., Zoo Biol, 37 (2), 98-106. 2018.” in line 40, 43, 103, and 118.

Comment 3: Can you give us the nutrient concentrations of their diet? How much AA, DHA and EPA are they actually ingesting? As it stands, all your results support is that there is inflammation in animals with diarrhea. You need the diet information to link with AA. If they eat the same diet and same amount of fatty acids, then causation will be somewhere else. You need the diet information to tie this all together or change the scope of your paper to inflammation and diarrhea.

<Response>

We newly added below table as Table 1, describing the concentration of each PUFAs contained in diet. As shown in table, sum of the concentration of n-6 and n-3 PUFAs were comparable in each diet fed marmoset in three institutions. In addition, we fed the same diet to healthy and MWS marmoset in each institution. Thus, the changes in urinary lipid profiles may reflect some inflammation but not the variation of the concentration of PUFAs derived from diet. We added Table 1 and related descriptions in line 90-92 and 161-162.

 PUFAs (g/100 g food) KUPRI Jikei CLEA

n-6 Linoleic acid 4.16 (1.66-2.82) 3.12 (2.5) 4.70 (1.88)

 Arachidonic acid N.D. 0.04 (0.03) N.D.

n-3 Linolenic acid 0.24 (0.10-0.32) 0.31 (0.25) 0.54 (0.21)

 DHA 0.11 (0.02-0.05) 0.35 (0.28) 0.04 (0.02)

 EPA 0.08 (0.01-0.03) 0.02 (0.01)

Reviewer #3: I commend the author's efforts to try and shed light on to disease like WMS which remains a mystery even after so many attempts, however I do think there are some major flaws in the design. I do understand that sample sizes can be challenging but in a prevalent colony as described by the owners, a n of 7 for both groups do significantly limit the conclusions that can be drawn. The title says production profile - I would remove production. I do think there are many authors, actually more authors than study subjects and it would be good to know if they satisfied the criteria for authorship.

<Response>

This study was the first trial of focusing on lipid metabolism in MWS marmosets. We understood the limitation of this study by small sample size so that we do not try to conclude the cause of MWS. But instead, we proposed the possibility of the importance of AA in MWS. We believe that it is important to shed light on MWS from various aspects.

As reviewer suggested we removed the “production” from title. And some author did not satisfy the criteria for authorship, their name is listed in Acknowledgement.

Comment 1: Ln24: It is a syndrome and hence more symptoms are needed to justify the syndrome.

<Response> 

We added the maximum weight of each marmoset in Table 2. Indeed, they all had persistent weight loss for over six months. We added the section on Identification of MWS in line 102 Methods Urine Samples.

Comment 2: Ln 34: I do not think given the low number of subjects, a substantial conclusion can be got from this.

<Response> 

We understand the limitation of this study by small sample size. We added the descriptions indicating that we did not reveal the mechanisms of MWS in this study as in Abstract and line 257 as following.

Abstract; “In this study, we found some urinary lipid metabolites suspected to be involved in the development of MWS. Although the cause of MWS remains unclear, our findings may provide some insight into understanding the mechanisms of development of MWS.” line 257; “We found some lipid metabolites which may be involved in the development of MWS. Since MWS is multifactorial disease and the cause of it remains unclear, we need further investigation to reveal a causal relationship. We believe our findings may provide some insight into understanding the mechanisms of development of MWS.”

Comment 3: Ln 42: I do think it is redundant to have steroid and glucocorticoids in the same line

<Response> 

We deleted the “steroid” in line 42.

Comment 4: Ln 72: Sampling method should be elaborated. LN 104; Were there liners? How did you clean them in between. If feces and urine fall on the same surface, how does one confirm there is no residual effect of the feces. How do you know these inflammatory lipids were not from a biofilm on the floor? The trays need to be described?

<Response> 

As we described above as response to the reviewer #1 comment 2, some of the control marmosets were in pair or family cages, while others were in individual cages for research or management purpose (not related to this study). MWS marmosets were in individual cages when their conditions were deteriorating and required intensive care. When the marmoset was in a pair or family cage, the marmoset was temporarily separated from other individuals using separation wall to collect urine samples. In some occasions, even in a pair or family cage, when a marmoset was urinated in front of us, the urine sample was collected immediately. A clean dry tray was placed under the cage. All urine samples were collected only when visually confirmed not to be contaminated with feces, another individual’s urine, or dropped food. After each sample collection, the trays were sterilized, rinsed thoroughly and dried. The urine samples were stored until use at -80°C. We added the descriptions in line 116.

Comment 5: Ln 84 onwards: Diet is a huge component of serum and urine lipid profiles. They should have looked at animals on the same diet to see how they compared.

<Response> We newly added below table as Table 1, describing the concentration of each PUFAs contained in diet. As shown in table, sum of the concentration of n-6 and n-3 PUFAs were comparable in each diet fed marmoset in three institutions. In addition, we fed the same diet to healthy and MWS marmoset in each institution. Thus, the changes in urinary lipid profiles may reflect some inflammation but not the variation of the concentration of PUFAs derived from diet. We added Table 1 and related description in line 90-92 and 161-162.

 PUFAs (g/100 g food) KUPRI Jikei CLEA

n-6 Linoleic acid 4.16 (1.66-2.82) 3.12 (2.5) 4.70 (1.88)

 Arachidonic acid N.D. 0.04 (0.03) N.D.

n-3 Linolenic acid 0.24 (0.10-0.32) 0.31 (0.25) 0.54 (0.21)

 DHA 0.11 (0.02-0.05) 0.35 (0.28) 0.04 (0.02)

 EPA 0.08 (0.01-0.03) 0.02 (0.01)

Comment 6: Ln 104: Urines should be replaced with urine samples. 

We revised. Thank you.

Comment 7: Ln 104: What was the duration between voiding and urine collection and storage. Volatile lipids can potentially be lost. Marmosets in certain facilities are trained to urinate, was an attempt for this made.

<Response>

Thank you for your advices. As descried in Response 4, the urine samples were collected as soon as they were found and only when they were not contaminated with feces, urine or dropped food. And then, the urine samples were stored until use at -80°C. We revised Methods Urine samples section in line 116.

Comment 8: Ln 111 onwards: What was the normalization technique used? Was the quantity of metabolites normalized to USG or creatinine. There should also be some way to look into if there was active sediment or not, or at least clinical signs or something to substantiate that these mediators were not produced within the urinary tract as a result of inflammation or infection. WMS is syndrome and hence this is important.

<Response>

We corrected the vales of lipid metabolites by creatinine concentration as described in line 149. Before the collection, we confirmed MWS based on the criteria (Cabana et al 2018) including the absence of pathogenic enteric bacteria and endoparasites. 

Comment 9: ln 146: This is an over stretch and this conclusion is not substantiated. I do agree with the other reviewers that the CV and the study design and conclusions have severe limitations, this along with small number of marmosets make me question the validity of these findings. While I do understand how difficult it is to work with marmosets, and samples can be scarce, I do not think the current study can be published as is in good faith. Sorry.

<Response> 

We understood the limitation of this study with sample size. This study was the first trial of focusing on lipid metabolism in MWS marmosets. We do not try to conclude that this is the cause of MWS, but instead, we proposed the possibility of the importance of AA in MWS. As described in Response to comment 4, we added the descriptions in Abstract and line 257.

Comment 10: 182- urines should be replaced with urine samples.

<Response>

We revised. Thank you.

---

## [Decision Letter · Decision Letter 1]

5 May 2020

PONE-D-20-02190R1

The profile of lipid metabolites in urine of marmoset with wasting syndrome

PLOS ONE

Dear Prof. Murata,

Thank you for submitting your manuscript to PLOS ONE. After careful consideration, we feel that it has merit but does not fully meet PLOS ONE’s publication criteria as it currently stands. Therefore, we invite you to submit a revised version of the manuscript that addresses the points raised during the review process.

We would appreciate receiving your revised manuscript by Jun 19 2020 11:59PM. To enhance the reproducibility of your results, we recommend that if applicable you deposit your laboratory protocols in protocols.io, where a protocol can be assigned its own identifier (DOI) such that it can be cited independently in the future. For instructions see: http://journals.plos.org/plosone/s/submission-guidelines#loc-laboratory-protocols

We look forward to receiving your revised manuscript.

Kind regards,

Petr Heneberg

Academic Editor

PLOS ONE

Reviewers' comments:

Reviewer's Responses to Questions

**Comments to the Author**

1. If the authors have adequately addressed your comments raised in a previous round of review and you feel that this manuscript is now acceptable for publication, you may indicate that here to bypass the “Comments to the Author” section, enter your conflict of interest statement in the “Confidential to Editor” section, and submit your "Accept" recommendation.

Reviewer #1: All comments have been addressed

Reviewer #3: (No Response)

2. Is the manuscript technically sound, and do the data support the conclusions?

Reviewer #1: Yes

Reviewer #3: Yes

3. Has the statistical analysis been performed appropriately and rigorously? 

Reviewer #1: Yes

Reviewer #3: Yes

4. Have the authors made all data underlying the findings in their manuscript fully available?

Reviewer #1: Yes

Reviewer #3: Yes

5. Is the manuscript presented in an intelligible fashion and written in standard English?

Reviewer #1: Yes

Reviewer #3: Yes

6. Review Comments to the Author

Reviewer #1: In my opinion the authors improved the text according to the questions raised by the reviewers. It is acceptable for publication.

Reviewer #3: Good job on the revisions. Thank you.

I do thing this needs some language editing and proof checking.

Please check the manuscript for language errors.

- 45 MWS. There is also no definitive diagnosis method for MWS.

insert 'antemortem'

89 Vitamin D. In CLEA, they were fed on 40-50 g CMS-1M (CLEA japan,

- J should be uppercase in Japan

95 such as fecal condition, appetite, hair condition and movement, and if they were suspected *any

96 problems, they would be **consulted with veterinarians.

- Add *to have, ** evaluated by the veterinarian

105 and/or antibiotics, and persistent weight loss [2]. The veterinarians in each institution assessed the

106 animals and *included in this study.

-*Prior to inclusion in the study

157 We comprehensively measured

- delete comprehensively

257 *multifactorial disease and the cause of it remains unclear,

- please add a

In all of the below line numbers, I do think they should say, marmosets with MWS, rather than MWS

109 however, that of MWS were around or less than 300 g at urine samples collection. Frequent and

110 recurrent diarrhea, and persistent weight loss for over six months were observed in MWS

116 as healthy (N = 7) in primary care or MWS (N = 7).

172 acid (DHA) in heathy (N = 7) and MWS (N = 7)

175 The urinary levels of arachidonic acid (AA)-derived catalyzed metabolites in heathy (N = 7) and

176 MWS (N = 7). di. dihydro; te, tetranor; COX, Cyclooxygenase; CYP, CYP450 epoxygenase; LOX,

177 lipoxygenase. *p<0.05, **p<0.01 compared to healthy urines.

180 The urinary levels of arachidonic acid (AA)-derived isoprostanes produced by non-enzymatic

181 oxidation in healthy (N = 7) and MWS (N = 7). **p<0.01 compared to healthy urines.

186 metabolite in healthy (N = 7) and MWS (N = 7). *p<0.05 compared to healthy urines. LOX,

187 lipoxygenase; COX, cyclooxygenase; CYP, cytochrome P450.

218 urinary lipid metabolites in MWS. These results may reflect that MWS marmosets may have

222 in plasma, 13,14-dihydro-15-keto derivatives, were increased in the urines of MWS [22]

231 amounts of PGE2, PGF2α and a metabolite or TXA2 were markedly increased in the urines of MWS.

234 ameliorated dextran sodium sulfate–induced colitis in mice [26]. Collins D et al. showed in an in vitro

240 In this study, we found that F2-IsoPs were increased in the MWS urine samples

250 this study, only 5 types of n-3 PUFAs-derived metabolites were constantly detected in MWS urines

The abstract has many language errors

7. PLOS authors have the option to publish the peer review history of their article (what does this mean?). If published, this will include your full peer review and any attached files.

Reviewer #1: Yes: Jerzy Juskiewicz

Reviewer #3: Yes: Joseph Cyrus Parambeth

---

## [Author Response · Author response to Decision Letter 1]

28 May 2020

The production profile of lipid metabolites in urine of marmoset wasting syndrome. 

Review Comments to the Author

Reviewer #1: In my opinion the authors improved the text according to the questions raised by the reviewers. It is acceptable for publication.

<Comments>

Thank you very much for your time and suggestions.

Reviewer #3: Good job on the revisions. Thank you. I do thing this needs some language editing and proof checking. Please check the manuscript for language errors.

<Comments>

Line 45 MWS. There is also no definitive diagnosis method for MWS. insert 'antemortem'

Line 89 Vitamin D. In CLEA, they were fed on 40-50 g CMS-1M (CLEA japan, - J should be uppercase in Japan

Line 95-96 such as fecal condition, appetite, hair condition and movement, and if they were suspected *any problems, they would be **consulted with veterinarians. - Add *to have, ** evaluated by the veterinarian

Line 105-106 and/or antibiotics, and persistent weight loss [2]. The veterinarians in each institution assessed the animals and *included in this study. - *Prior to inclusion in the study

Line 157 We comprehensively measured - delete comprehensively

Line 257 *multifactorial disease and the cause of it remains unclear, - please add a

In all of the below line numbers, I do think they should say, marmosets with MWS, rather than MWS

Line 109 however, that of MWS were around or less than 300 g at urine samples collection. Frequent and recurrent diarrhea, and persistent weight loss for over six months were observed in MWS

Line 116 as healthy (N = 7) in primary care or MWS (N = 7).

Line 172 acid (DHA) in heathy (N = 7) and MWS (N = 7)

Line 175 The urinary levels of arachidonic acid (AA)-derived catalyzed metabolites in heathy (N = 7) and MWS (N = 7). di. dihydro; te, tetranor; COX, Cyclooxygenase; CYP, CYP450 epoxygenase; LOX, lipoxygenase. *p<0.05, **p<0.01 compared to healthy urines.

Line 180 The urinary levels of arachidonic acid (AA)-derived isoprostanes produced by non-enzymatic oxidation in healthy (N = 7) and MWS (N = 7). **p<0.01 compared to healthy urines.

Line 186 metabolite in healthy (N = 7) and MWS (N = 7). *p<0.05 compared to healthy urines. LOX, lipoxygenase; COX, cyclooxygenase; CYP, cytochrome P450.

Line 218 urinary lipid metabolites in MWS. These results may reflect that MWS marmosets may have

Line 222 in plasma, 13,14-dihydro-15-keto derivatives, were increased in the urines of MWS [22]

Line 231 amounts of PGE2, PGF2α and a metabolite or TXA2 were markedly increased in the urines of MWS.

Line 234 ameliorated dextran sodium sulfate–induced colitis in mice [26]. Collins D et al. showed in an in vitro

Line 240 In this study, we found that F2-IsoPs were increased in the MWS urine samples

Line 250 this study, only 5 types of n-3 PUFAs-derived metabolites were constantly detected in MWS urines

The abstract has many language errors

<Responses>

Thank you for your kind checking our language errors. We carefully revised the errors thoroughly manuscript including abstract.

---

## [Editor Report · Decision Letter 2]

1 Jun 2020

The profile of lipid metabolites in urine of marmoset wasting syndrome

PONE-D-20-02190R2

Dear Dr. Murata,

We are pleased to inform you that your manuscript has been judged scientifically suitable for publication and will be formally accepted for publication once it complies with all outstanding technical requirements.

With kind regards,

Petr Heneberg

Academic Editor

PLOS ONE

---

## [Editor Report · Acceptance letter]

4 Jun 2020

PONE-D-20-02190R2 

The profile of lipid metabolites in urine of marmoset wasting syndrome 

Dear Dr. Murata:

I'm pleased to inform you that your manuscript has been deemed suitable for publication in PLOS ONE. Congratulations! Your manuscript is now with our production department. 

Kind regards, 

on behalf of

Dr. Petr Heneberg 

Academic Editor

PLOS ONE